# Impact of Oil-in-Water Adjuvanted β-Glucan on Innate Immune Memory in Piglets

**DOI:** 10.3390/vaccines12090982

**Published:** 2024-08-29

**Authors:** Razieh Ardali, Obdulio Garcia-Nicolas, Catherine Ollagnier, José María Sánchez Carvajal, Maria Levy, Pauline Yvernault, Francisco de Aboim Borges Fialho de Brito, Artur Summerfield

**Affiliations:** 1Institute of Virology and Immunology, 3147 Mittelhäusern, Switzerland; 2Department of Infectious Diseases and Pathobiology, Vetsuisse Faculty, University of Bern, 3012 Bern, Switzerland; 3Graduate School for Cellular and Biomedical Sciences, University of Bern, 3012 Bern, Switzerland; 4Swine Research Unit, Agroscope, 1725 Posieux, Switzerland; 5Department of Anatomy and Comparative Pathology and Toxicology, Faculty of Veterinary Medicine, University of Córdoba, 14014 Córdoba, Spain

**Keywords:** β-glucan, adjuvant, innate immune memory, innate tolerance, piglets

## Abstract

The non-specific protective effects offered by the concept of “innate immune memory” might represent a promising strategy to tackle early-life threatening infections. Here we tested the potential of an in vitro selected β-glucan in inducing trained immunity using an in vivo porcine model. We assessed the leukocyte transcriptome using blood transcriptomic module (BTM), proinflammatory cytokines, and clinical scoring after a first “training” and a second “stimulation” phase. The possible induction of innate immune memory was tested during a “stimulation” by an LPS-adjuvanted *Mycoplasma hyopneumoniae* vaccine (Hyogen^®^) one day after weaning. Following the “training”, no major group differences were found, with the exception of a plasma TNF that was only induced by Adj and Adj_BG treatment. After vaccination, all groups developed similar antibody responses. A significant induction of plasma TNF and IL-1β was found in groups that received Adj and Adj_BG. However, following vaccination, the expected early innate BTMs were only induced by the PBS group. In conclusion, the adjuvant alone, adjuvant-formulated β-glucan, or orally applied β-glucan were unable to enhance innate immune reactivity but rather appeared to promote innate immune tolerance. Such an immune status could have both positive and negative implications during this phase of the piglet’s life.

## 1. Introduction

Birth and weaning are critical transition phases during early life with heightened vulnerability to infections. Exposure of the naïve and immature immune system to a wide range of pathogens and other environmental stress factors is among the main challenges resulting in early-life threatening infections [1,2]. The naïve immune system of the neonate imposes a greater reliance on innate immune responses for controlling pathogenic infection throughout this period [3,4]. In the last few years, several studies have revealed that exposure to certain pathogen or damage-associated molecular patterns (PAMPs and DAMPs) induces durable epigenetic and metabolic alterations in prototypical innate immune cells such as monocytes, macrophages, and natural killer cells. These modifications can either potentiate or impair the response of innate cells to restimulation with heterogenous or homogenous insults, a phenomenon referred to as “trained immunity” or “tolerance”, respectively [5,6,7,8]. Collectively, these enhancing or suppressive effects on innate immunity have also been termed “innate immune memory” [9]. In this regard, the Bacille Calmette-Guérin (BCG) vaccine, widely administered early after birth, has been associated with reduced mortality in neonates by providing protection against non-specific pathogens [10]. These off-target effects of BCG are partly mediated by epigenetically reprogrammed innate immune cells and their progenitors [11,12]. For instance, profiling of the DNA-methylation pattern of monocytes isolated 14 months after BCG vaccination in neonates has revealed a long-lasting DNA methylation signature enriched for terms related to innate immunity [13]. The non-specific protective effects observed with several vaccines or their accompanying adjuvants have also been linked to mechanisms mediated by trained immunity [14,15]. Interestingly, studies have shown that innate immune memory induced by sublethal systemic infection with *Candida albicans* or zymosan can be transmitted to offspring, providing protection against non-specific challenges [16]. These observations have led to the idea that prophylactic administration of immunomodulatory agents early in life can stimulate host protective immunity. For instance, provoking innate immunity by toll-like receptor 4 (TLR4) or TLR7/8 agonists has been shown to enhance survival to polymicrobial sepsis in mice during neonatal life [17]. The immunomodulatory effects of β-glucans, broadly studied in the context of trained immunity, have rendered them attractive candidates for enhancing immune responses to pathogens and developing vaccine adjuvants or vaccine delivery systems [18,19,20,21,22]. The training effects of β-glucans have been attributed to their potential to trigger signaling pathways downstream of dectin-1. Indeed, activation of the mammalian target of rapamycin (mTOR) via the dectin-1–AKT–hypoxia-inducible factor 1α (HIF-1α) axis has been reported to be associated with the metabolic and epigenetic modifications required for the induction and maintenance of β-glucan-induced trained immunity [23]. Several studies have addressed the protective effects of prolonged supplementation of the diet with β-glucans against post-weaning complications in piglets [24,25,26,27]. 

In the current study, we have focused on the transcriptional response induced by schizophyllan, a non-ionic, water-soluble β-glucan derived from *Schizophyllum commune*, formulated in an oil-in-water adjuvant and administered via the i.m. route to piglets at 14 days of age. To evaluate the capacity of this β-glucan to induce innate immune memory, we investigated the transcriptomic signature of a single dose of β-glucan and its ability to alter immunoreactive characteristics of piglets to an LPS-adjuvanted vaccine against *M. hyopneumoniae* (Hyogen^®^). 

## 2. Material and Methods

### 2.1. In Vitro Assessment of the Immunomodulatory Effects of Various β-Glucans

Blood samples were obtained from 14- to 18-month-old Large White pigs kept under specific pathogen-free (SPF) conditions at the Institute of Virology and Immunology (IVI, Mittelhäusern, Switzerland). Peripheral blood mononuclear cells (PBMCs) were isolated using density gradient centrifugation (Ficoll-Paque™ PLUS (1.077 g/L; GE Healthcare Life Sciences, Dübendorf, Switzerland). To isolate monocytes, PBMCs were then incubated with CD172a monoclonal antibody (mAb, clone 74-22-15A; hybridoma kindly provided by Prof. Armin Saalmüller, Veterinary University of Vienna, Vienna) followed by washing and incubation with mouse IgG beads (Miltenyi Biotech, Bergisch Gladbach, Germany) and sorting by LS MACS columns (Miltenyi Biotec). Monocytes then were cultured at the density of 25 × 10^4^/mL in Dulbecco’s modified Eagle’s medium (DMEM; Thermo Fisher Scientific, Waltham, MA, USA) supplemented with 10% fetal bovine serum (FBS) and porcine macrophage-colony stimulating factor (mCSF, 100 U/mL) [28] in 48-well plates and incubated at 39 °C with 5% CO_2_ [28]. Freshly isolated monocytes were stimulated overnight with different β-glucans including scleroglucan (10 μg/mL; Invivogen, San Diego, CA, USA), β-glucan peptide (10 μg/mL; Invivogen, San Diego, CA, USA), pustulan (10 μg/mL; Invivogen, San Diego, CA, USA), curdlan (10 μg/mL; Invivogen, San Diego, CA, USA), and Schizophyllan (10 μg/mL; BOC Science, Shirley, NY, USA). Then the supernatants were collected, and the cells were washed with warm PBS and then incubated for 5 days in the above medium, which was refreshed after 3 days. After 6 days, the cells were restimulated with 1 µg/mL of LPS (Sigma-Aldrich, Buchs, Switzerland). for 24 h and the supernatants were collected and stored at −20 °C for further analysis. 

In addition, monocyte-derived macrophages (MDMs) were differentiated from monocytes by culture in the above medium and then stimulated with either Schizophyllan or LPS (10 ng/mL). After 24 h of MDM stimulation, supernatants were collected for cytokine quantification. 

### 2.2. Ethics and Animal Experimental Conditions

The experiments in pigs were conducted in compliance with the animal welfare regulations of Switzerland (TSchG SR 455; TSchV SR 455.1; TVV SR 455.163). The committee on animal experiments of the canton of Fribourg, Switzerland reviewed the pig experimentation and pig blood collection protocols, and the cantonal veterinary authorities (Amt für Lebensmittelsicherheit und Veterinärwesen, Givisiez, Switzerland) approved the animal experiments. Twenty-four healthy piglets (male and female, >3.5 kg, 14 (±1) day of age) from three litters of the Agroscope herd were included in this study. The experimental farm was free from any *M. pneumoniae* infection. To avoid litter, weight, and gender effects, piglets in the three litters were equally and randomly assigned to the four treatment groups (see below), ensuring a balanced gender and weight distribution of the groups. Despite the assignment, piglets remained with their mother sow until weaning (25 days of age) and were then kept in the same pen for two weeks. At 39 days of age, they were regrouped in pens of 12 animals that were equally mixed from the three litters as well as the treatment groups. The housing facility was authorized by the Food Safety and Veterinary Office LSVW/SAAV of the Canton of Fribourg under authorization # H-08.2014-FR and complied with animal welfare requirements. Eye and snout contacts were possible between pens. Thermic comfort (≥20 °C) was ensured by measuring the room temperature and providing an additional heat source (infrared lamp, radiant heater) if needed. Food and water were available ad libitum through feeders and nipple drinkers. The feed consumption was monitored by weighing once daily the amount of food delivered and the left-overs.

### 2.3. Treatment Groups

Three groups (six piglets per group) received 1 mL of either PBS (control group), adjuvant (Adj; Montanide 28R, Seppic), or β-glucan (Schizophyllan) formulated in adjuvant (Adj_BG) via i.m. injection in the neck area at day 14 of age. For the adjuvant, 1500 mg Montanide 28R (kindly obtained from Andra Corder, Seppic, Castre, France) was mixed with 8500 mg sterile H_2_O and stirred for 10 min at 1000 rpm at room temperature using IKA Ultra Turrax^®^ Tube Drive mixer. For the Adj_BG group, 2.5 mg β-glucan (50 μg/kg) was dissolved in 8500 mg sterile H_2_O and mixed with 1500 mg Montanide 28R as for the Adj group. The fourth group received daily freshly mixed Macrogard (Orffa, Breda, NL; OBG) at 50 mg/kg with 2 mL of full-fat milk via the oral route for the duration of 10 days. The rationale for including the OBG group was based on previous reports demonstrating that oral administration of Macrogard was able to enhance protective immunity against *E. coli* [26]. Piglets were weaned at 28 days of age, and on the following day received a 2 mL i.m. injection of an LPS-adjuvanted vaccine against *M. hyopneumoniae* (Hyogen^®^, Ceva) (Figure 2A). Body temperature, weight, and the injection site were monitored before and after both stimulation and vaccination for five consecutive days. The injection site reactions were monitored using a scoring system explained in Table 1. Blood samples were collected at 0, 1, 3, 7, and 14 days post-stimulation and 0, 1, 7, 14, 21, and 28 days post-vaccination. Blood collections consisted of 2 mL volumes on D0, D1_PS, and D3_PS, and 4 mL for the rest of the experiment, except during slaughter, when 50 mL of blood was collected.

### 2.4. Blood Sample Preparation

For plasma, blood samples were centrifuged for 10 min at 500× *g* using a refrigerated centrifuge, and plasma aliquots were stored at −20 °C. For leukocyte isolation, the blood was transferred to a 50 mL polypropylene tube containing 16 mL of prewarmed (37 °C) 1× red blood cell ammonium chloride lysis buffer (0.15 M NH_4_Cl, 10 mM NaHCO_3_, 1 mM EDTA) and left for 180 s to allow red blood lysis. Then the tubes were filled to the maximum with 4 °C cold PBS and centrifuged at 500× *g* for 5 min at 4 °C. The supernatant was discarded, and the pellet was re-suspended in 50 mL PBS (4 °C) followed by centrifugation at 300× *g* for 5 min at 4 °C. Next, the cell pellets were re-suspended in 1500 µL of Trizol (Thermo Fisher) and the samples were stored at −80 °C. 

### 2.5. Cytokine Measurements

The levels of TNF, IL-6, IL-1β, IL-8, IL-10, and IL-1RA in plasma samples were determined using commercial ELISA kits (R&D Systems). 

### 2.6. Antibody Measurements 

The presence of antibody against *M. hyopneumoniae* in plasma samples was assessed using IDEXX Mhyo Antibody ELISA Kit (IDEXX Europa B.V., Hoofddorp, Netherlands). Briefly, the samples were diluted 1/40 by mixing 10 µL of plasma with 390 µL of sample diluent. Samples were transferred to antigen-coated 96-well plates at 100 µL/well. After incubation for 30 min at room temperature, the plate was washed three times, and anti-porcine Ig horseradish peroxidase conjugate was added for 30 min. After a wash step, 100 µL 3,3′,5,5′-tetramethylbenzidin substrate was added for 15 min, followed by adding the stop solution. Optical density (OD) values were measured using the Tecan Sunrise reader (Tecan Group LTD, Männedorf, Switzerland) at a wavelength of 650 nm [29]. 

### 2.7. Bulk-RNA Sequencing and Analysis

RNA was extracted using the Nucleospin RNA kit (Macherey Nagel). The RNA concentration was quantified using the Qubit RNA HS assay kit and its integrity was determined by fragment analyzer (5200 Fragment Analyzer CE instrument, Agilent). Libraries were prepared with the BRB-seq protocol followed by sequencing with an Illumina^®^ NovaSeq6000 sequencer (Illumina). The quality of reads was evaluated by FastQC v.0.11.21. Reads were mapped to the sus scrofa v.11.1 genome and Ensembl annotation version 1.108 using STAR version 2.7.10b. The 249 sequenced libraries had an average size of 3,969,430 reads (median: 3,577,174, min: 811,155, max: 10,337,503). Differential gene expression analysis (DEA) was performed using DESeq2 v. 1.36 after batch correction [30]. A pre-filtering before the DEA analysis removed genes with ≤10 reads in ≤5 libraries, keeping 11,605 out of a total of 31,908 annotated genes. The genes were ranked based on the “stat” value and subjected to “ranked gene set enrichment analysis” (GSEA) using the GSEA v_4.3.2 and porcine blood transcriptomic module (BTM) gene sets [31]. Three hundred forty-seven BTMs were included which were related to immune cell population distribution, cellular processes such as cell cycle, transcription, leukocyte-specific signaling pathways, leukocyte migration, activation of particular immune cell types such as dendritic cells and T cells, inflammation, coagulation, platelet activation, antiviral responses, antigen presentation, immunoglobulin production, or metabolic processes relevant for immune responses [32,33,34]. Significant BTMs (false discovery rate (FDR) < 0.05) were visualized using the ggplot2 package in R. 

### 2.8. Statistical Analysis

Data analysis and figures were done using R version 4.2.0 and GraphPad Prism 10 Software (GraphPad Software). Body temperature was analyzed using a linear mixed-effects model to compare each group with its baseline (D0) and to compare the differences between groups at each time point. The Kruskal–Wallis test was used for comparing the injection site score between groups at each time point, with Dunn’s correction for multiple comparisons. One-way ANOVA was used for comparing the cytokine levels between different groups at each time point, with Tukey’s correction for multiple comparisons. Multiple paired t-tests were used to compare the cytokine levels within each group against the baseline with Bonferroni correction for multiple comparisons. The antibody levels were analyzed using a linear mixed-effects model. *p* values lower than 0.05 were considered as statistically significant. 

## 3. Results 

### 3.1. In Vitro Selection of Immunostimulatory β-Glucans

To select the β-glucan for the in vivo study, several β-glucans were tested on porcine monocytes and MDMs with respect to their ability to induce the proinflammatory cytokines TNF and IL-1β. In monocytes, scleroglucan, pustulan, and Schizophyllan were found to induce high levels of TNF (Figure 1A). Due to its solubility and thereby lower risk for induction of possible adverse local effects upon intramuscular injection, Schizophyllan was selected to be further tested for other cytokines and the stimulation of MDMs. Stimulation of monocytes with Schizophyllan led to the induction of IL-1β comparable to LPS (Figure 1B). Also, the stimulation of MDMs with Schizophyllan-induced TNF and IL-1β similar to LPS (Figure 1E,F). 

All selected β-glucans were also tested for the ability to train monocytes for enhanced cytokine responses of MDM to second stimulation with LPS. Unfortunately, none of the β-glucans tested caused increased responses of MDM in terms of TNF. Shizophyllan was also tested for IL-1β and found to also suppress the secondary responses to LPS for this cytokine. In all cases, the medium values of cytokines were even reduced and this was statistically significant for curdlan and schizophyllan (Figure 1C,D). 

Based on its ability to induce innate activation of both monocytes and macrophages and its solubility, we selected Schizophyllan for further in vivo evaluation. 

### 3.2. Safety Parameters 

We first evaluated safety parameters following injection of adjuvant alone (group Adj), adjuvant with schizophyllan (group Adj_BG), PBS, or after oral treatment of pigs with Macrogard (OBG). None of these treatments significantly enhanced the body temperature, when compared to PBS injection (Figure 2B). In contrast, 8 h after vaccination, an increase in body temperature was observed in all groups, which was significant compared to the baseline for each group. However, there was no significant difference between groups at this time point (Figure 2C). For the injection site score, we observed a significant increase in the Adj_BG group. This lasted for 72 h (Figure 2D). After vaccination, only two piglets showed reactions, with no differences between the groups (Figure 2E). The body weights increased over time with no differences between the groups (Figure 2F). 

### 3.3. Cytokine and Antibody Profiles

Following stimulation, Adj and Adj_BG-treated piglets showed significantly increased levels of TNF at day 1 post-stimulation (D1_PS) compared to day 0 (Figure 3A). Furthermore, group comparisons post-stimulation demonstrated higher levels of TNF in the Adj compared to the PBS group. In none of the treatment groups did the induction of IL-1β and IL-1RA show a significant difference in the plasma (Figure 3B,C). Also, there was a significant induction of IL-10 in Adj_BG-treated piglets at D1_PS compared to day 0 (Figure 3D). However, no significant change was detected in the group comparison at D1_PS for this cytokine. 

After vaccination, enhanced TNF levels were found in the plasma at 24 h in PBS, Adj, and Adj_BG-treated piglets relative to their baseline (Figure 3E). For IL-1β, Adj and Adj_BG showed significant increases compared to their baseline (Figure 3F). For both TNF and IL-1β, no significant changes were observed relative to PBS-treated piglets. For IL-1RA, the induction was limited to only one animal in Adj-treated piglets and several PBS-treated animals (Figure 3G). For IL-10, only Adj_BG-treated animals showed a significant increase compared to their baseline (D14_PS), without any significant difference in the level of group comparison (Figure 3H). IL-6 and IL-8 were neither detected after stimulation nor after vaccination. 

We also measured the induction of antibody against *M. hyopneumoniae* at days 14 and 21 post-vaccination. The within-group comparison showed a significant induction of antibody in OBG-treated animals on day 21 PV compared to day 14 PV (Figure 3I). However, there was no significant difference between groups at both time points. 

These first results indicated that the adjuvant induced an innate immune response that was not further enhanced by the addition of β-glucan. Also, the vaccine induced an innate response in PBS, Adj, and Adj_BG-treated animals in terms of TNF, and in Adj and Adj_BG groups in terms of IL-1β. 

### 3.4. Transcriptomic Profile of Leukocytes Post-Stimulation

To better understand innate immune responses, blood leukocytes collected at various time points post-stimulation (PS; D0, D1_PS, D3_PS, D7_PS, D14_PS) and post-vaccination (PV; D1_PV, D7_PV) were transcriptionally profiled using the BTM analysis, which we have used in many vaccination and infection studies [31,32,33,35]. The BTM platform informs on changes in immune cell population distribution, cellular processes, such as cell cycle and transcription, leukocyte-specific signaling pathways, leukocyte migration, activation of immune cell types such as dendritic cells, monocytes, B and T cells, inflammation, coagulation, platelet activation, antiviral responses, antigen presentation, immunoglobulin production, and on metabolic processes.

For these analyses, differentially expressed genes (DEGs) were calculated by pairwise comparison of all time points to the baseline values. For post-stimulation time points, the comparisons were performed relative to the D0 baseline. For the post-vaccine phase, the baseline was the pre-vaccination time point (D14_PS). The number of DEGs for each of these comparisons is shown in Table 2. In all groups including the PBS group, a high number of DEGs were found for all time points. This indicates that many changes are not related to treatment but to piglet development and unknown environmental factors. This made further interpretation of the DEGs difficult and we decided to focus on the BTM analyses which include only ca. 4000 immunologically relevant genes. Based on previous work by us and others using adjuvanted vaccines, an early induction of many innate BTMs between D1 and D3, which is typically followed by a downregulation of the same BTM, is expected [31,36]. 

For the innate immune system BTM (Figure 4A), at day 1 post-stimulation (D1_PS), only one module (“enriched in monocyte” M11.0) showed positive enrichment for Adj, and two BTMs, “TLR and inflammatory signaling” M16 and “monocyte surface signature” S4, were positively enriched for Adj_BG. In the OBG group, we found a downregulation of the inflammatory module “inflammasome receptors and signaling” (M53). At D3, the inflammatory BTM “proinflammatory dendritic cell, myeloid cell response” (M86.1) was induced in the Adj group and M16 was induced in the PBS group. For the remaining time points, the innate BTMs were mostly downregulated with no group-specific pattern being visible, and there were similar effects in the PBS controls. These data indicate that only a weak early innate BTM response was induced in the group that received adjuvants.

In terms of adaptive immune system BTMs (Figure 4B), previous data would lead us to expect an early downregulation of adaptive BTM associated with cell cycle and lymphocytes, followed by an upregulation at D7 [31]. In the Adj group, we found positive enrichment for several BTMs related to cell cycle between days 3 and 14 PS. In contrast, the Adj_BG only induced BTM directly related to the cell cycle by D14. Interestingly, the OBG group showed only negative enrichment for the cell cycle and T cell BTMs at all time points. This was also the only group showing a positive enrichment for two modules related to plasma cells at D3_PS. In the PBS group, we also identified an increase in a considerable number of cell cycle BTMs at D7 and 14, underlining the impact of pig development and environmental factors on BTM development. Altogether, while the data indicate treatment-dependent effects, their interpretation must take the dynamic developmental changes during this period of the piglet’s life into consideration. Nevertheless, together with the cytokine data, it can be concluded that Adj treatment induced a weak systemic innate immune and cell cycle response that appeared to be blunted by the addition of BG. A possible anti-inflammatory effect of BG was supported by the OBG treatment data. In this group, no signs of innate and adaptive immune activation, with the exception of a plasma cell response, were observed. 

### 3.5. Transcriptomic Profile of Leukocytes Post-Vaccination

In terms of innate immune BTMs, strikingly only the PBS group showed a strong positive enrichment for many innate BTMs including six antiviral, 12 dendritic cells (DC)-related, 12 inflammation, and 13 myeloid cells-related modules at D1_PV. Also, two NK cell-related modules showed an early negative enrichment only in the PBS group (Figure 5A). 

Regarding the adaptive BTMs (Figure 5B), again only the PBS group showed an early negative enrichment for 5 B- and 10 T cell-related modules at D1_PV. For the cell cycle BTM, very similar profiles were found with the Adj, Adj_BG, and OBG groups that showed negative enrichment for most BTMs at D1 and D7. At D7_PV, all the groups had a similar signature with no clear induction of adaptive BTMs, with the exception of plasma cell modules and “enriched in cell cycle” M167. Altogether, the transcriptomic data indicate that Adj, AdjBG, and OBG induce a tolerogenic innate immune memory. 

## 4. Discussion

Based on the well-established concept of innate immune memory, the current study tested the effect of β-glucan as well as adjuvant in 14-day-old piglets, focusing on its impact on in vivo plasma cytokine and leukocyte transcriptomics following a secondary stimulus which was an LPS-adjuvanted commercial vaccine. To this end, we injected a single dose of soluble β-glucan formulated in an oil-in-water adjuvant or the same adjuvant alone for training that was followed by a second stimulation with an LPS-adjuvanted vaccine against *M. hyopneumoniae* after a two-week interval. The rationale behind this study approach was as follows: firstly, the reported capacity of β-glucan and certain adjuvants in inducing long-lasting transcriptional and epigenetic modifications in myeloid cells and their progenitors [37,38]; and secondly, the role of the pre-vaccination immune state in shaping the vaccination outcome and antibody response. The latter comes from the studies on human subjects showing that a pre-existing proinflammatory signature in the blood transcriptional profile correlates with heightened antibody response across a variety of vaccines [39]. 

In terms of the immunomodulatory effects of β-glucan, the intramuscular injection of a single dose of adjuvant formulated β-glucan (Adj_BG), when compared to adjuvant alone, did not have a considerable impact on innate response in terms of innate BTMs. However, some animals in the Adj_BG group had injection site reactions that lasted for 72 h, suggesting the induction of local inflammation by β-glucan. In this line, soluble mannan derived from *Candida albicans* has been reported to be immunosilent in the periphery while inducing a potent inflammatory response in the draining lymph nodes (dLNs) [40]. 

Notably, the injection of Adj alone despite its induction of inflammatory cytokines did not lead to a significant immune perturbation in the level of transcriptomics. Assessing the transcriptional profile of several adjuvants including CAF01, IC31, GLA-SE, and Alum administered subcutaneously to mice, in isolated PBMCs and draining lymph nodes after 6, 24, 72 and 168 h, has shown distinct kinetics and magnitudes of responses for each adjuvant. Alum, the most commonly used adjuvant in human vaccines, and GLA-SE were reported to induce the smallest and largest changes in the transcriptional signature of both PBMCs and draining lymph nodes, respectively [41]. Despite subtle transcriptional changes induced by Alum, it serves as a potent adjuvant in mounting adaptive responses to vaccines [42,43]. 

Regarding the impact of stimulation/priming on response to vaccine, we did not observe any significant difference between Adj and Adj_BG in vaccine-induced proinflammatory TNF and IL-1β, although both groups induced a significant response relative to their baseline. Also, the antibody response against *M. hyopneumoniae* was not enhanced by any of the treatments. In contrast to our study, the injection of a single dose of β-glucan derived from *Euglena gracilis* 28 days prior to vaccination of dogs with rabies vaccine has been reported to induce a higher B cell response, determined by increased total rabies-specific IgG, rabies-specific plasma cells, and neutralizing antibodies [20]. 

Strikingly, at the transcriptomic level, we observed a strong innate signature in response to vaccine only in the PBS group. In terms of adaptive BTMs, while the PBS group showed downregulation of B and T cell-related modules at D1_PV, other groups did not induce this signature. Importantly, this signature of innate and adaptive BTM modulation was comparable to our previous data with pigs [31,36] and is also found with human vaccines [44]. Our transcriptomic data indicate that immunomodulation with Adj, Adj_BG, and OBG affected the early response to vaccination in a negative manner, possibly caused by a tolerogenic status in cells of the innate immune system. Such tolerogenic effects were also reported for AS03 adjuvant when it was included in an inactivated split-virion vaccine against H5N1 influenza. They showed that vaccination of human subjects with H5N1 + AS03 induced long-lasting innate refractoriness via reducing histone marks in monocytes. Long-lasting innate refractoriness was attributed to reduced chromatin accessibility at AP-1-targeted loci after vaccination. Despite this refractoriness signature on AP-1-targeted loci, the AS03 adjuvant led to elevated antiviral vigilance pronounced by enhanced IRF accessibility, IFN production, and protection against heterogeneous viral infection [44]. 

On the other hand, it was reported that priming with the CAF01 adjuvant enhanced the innate and adaptive BTMs after a booster immunization [45], highlighting differences between adjuvants in inducing innate immune memory. It is worth noting that, in these studies, adjuvants have been used in combination with the antigens and this might affect the behavior of adjuvants, or in other words, the immune response via the cross-talk between innate and adaptive immune systems. 

An interesting secondary observation was with respect to the administration of oral β-glucan. The transcriptomic profiling of these pigs after the 10-day treatment period demonstrated a negative regulation of all BTM with the exception of two plasma cell modules (M156.0 and M156.1), which was unique to this group. Also, the tolerogenic status following vaccine administration was remarkable. Given the potential impact of the oral administration on the microbiome and the mucosal immune system, alternative modes of action to the innate immune memory concepts might apply. Clearly understanding the mechanisms behind these observations is more complex but appears to be a very interesting topic for future investigations. 

Our study had some limitations that are important to consider for future work. The lack of apparent induction of trained immunity by β-glucan might be caused by a number of factors. (1) The route of administration has been reported to affect immunostimulant trafficking, local and systemic reactions, and innate and adaptive effectors [46]. In agreement with this, the induction of trained immunity in hematopoietic stem cells by intravenous BCG was not observed when it was administered subcutaneously [11]. (2) The formulation of β-glucan may not have been suitable to deliver the PAMP to its target cells and receptors. Also, the tolerogenic effects of the adjuvant might have negatively interacted with the β-glucan effects. Therefore, an adjusted formulation that is well suited to the physicochemical characteristics of immunostimulant might be required to achieve the desired response. For instance, the adsorption of mannan to aluminum hydroxide, which is facilitated by the presence of phosphate groups of mannans, can enhance local and systemic immune responses [40]. (3) The dose and source of β-glucans might not have been appropriate to induce innate immune memory. In this regard, while β-glucan isolated from *Candida albicans* induced trained immunity, laminarin has been reported to have anti-inflammatory characteristics [47,48]. (4) β-glucans might be unable to activate trained immunity in pigs. Indeed, we have screened a number of β-glucans in vitro and identified none that were able to enhance in vitro responses. This would also be in line with the work of others [49]. 

An additional limitation of our study was the high variability between animals in the groups, and also between the groups prior to stimulation. To avoid such effects, we recommend increasing the number of animals in future works or to use specific-pathogen-free pigs. Furthermore, any uncontrollable environmental effects such as infections and other stressors should be minimized. Our data demonstrate how dynamic changes are in the immune system of such young piglets. This is unavoidable but can make the interpretation of transcriptomic data difficult. Finally, additional readouts should be considered in future work. One possibility is assessing the ex vivo response of monocytes in cell culture combined with analyses of epigenetic changes and the frequency of the innate cells. 

Taken together, our results indicate that β-glucan at the dose, formulation, and route of administration used was unable to induce trained immunity. On the other hand, the oil-in-water adjuvant employed might induce reprogramming of innate cells toward a hyporesponsive or tolerant state. It is important to note that this was not associated with reduced vaccine responses and should not necessarily be considered negatively. However, negative non-specific effects appear to be a reality for certain human-inactivated vaccines while several live vaccines are associated with positive non-specific aspects [50]. Such effects should therefore be considered in adjuvant and immunomodulation research. 

## Figures and Tables

**Figure 1 vaccines-12-00982-f001:**
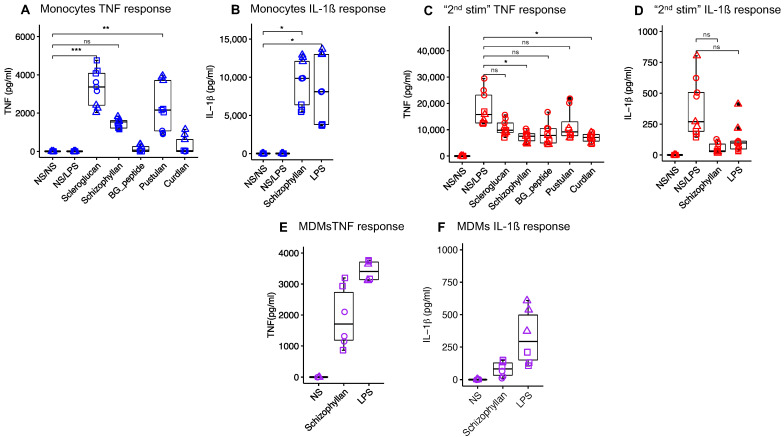
**Cytokine response of porcine monocytes and MDMs to different β-glucans.** Porcine monocytes were primed with different β-glucans (10 μg/mL) or LPS 10 ng/mL for 24 h. The stimuli were removed and the cells were rested for 6 days followed by a second stimulation with LPS (1 μg/mL) for 24 h. The levels of cytokines were measured after priming (**A**,**B**) and restimulation (**C**,**D**). NS/NS represents the cells that received neither the first nor the second stimulation. NS/LPS represents the cells that did not receive the first stimulation but received the second stimulation with LPS. (**E**,**F**) represent the cytokine response of porcine MDMs to stimulation with schizophyllan and LPS for 24 h. NS represents the MDMs that received only culture medium. Each shape represents a biological replicate. Comparison between groups was performed using One-way ANOVA (ns, not significant; * *p* < 0.05; ** *p* < 0.01; *** *p* < 0.001).

**Figure 2 vaccines-12-00982-f002:**
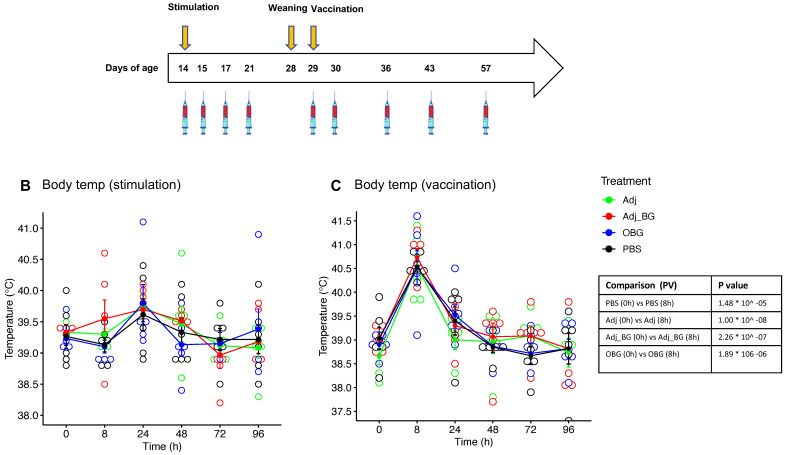
**Animal experiment (in vivo).** (**A**) Schematic representation of the in vivo study. (**B**,**C**) body temperatures after stimulation and vaccination. (**D**,**E**) injection site score after stimulation and vaccination. (**F**) the body weight during the experiment. Body temperatures were analyzed using a linear mixed-effects model for comparing each group with the baseline (D0) and comparing different groups at each time point. The injection site reaction was analyzed by the Kruskal–Wallis test. * *p* values ≤ 0.05 were considered as significant.

**Figure 3 vaccines-12-00982-f003:**
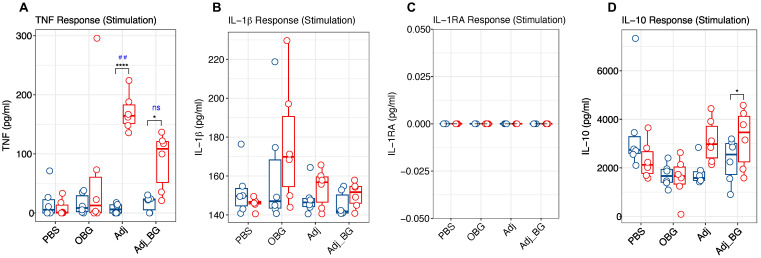
**Cytokine and antibody responses after stimulation and vaccination.** The levels of cytokines were measured in plasma samples collected at D0 and D1_PS for the post-stimulation phase and D14_PS and D1_PV for the post-vaccination phase. (**A**–**D**) TNF, IL-1β, IL-1RA, and IL-10 responses after stimulation. (**E**–**H**) TNF and IL-1β, IL-1RA and IL-10 responses after vaccination. (**I**). Antibody response at D14 and D21 post-vaccination was measured in plasma samples. Cytokine responses were analyzed using multiple paired t-tests for comparing each group with itself relative to the baseline and the Bonferroni test was used for multiple comparison corrections. One-way ANOVA was used for comparing groups with each other at each time point and Tukey was used for multiple comparison correction. Antibody response was analyzed by a linear mixed-effects model. Asterisk and “##” indicate the statistical significance for the comparison of each group to its baseline and comparing groups with each other at any time point, respectively. ns, not significant; ## *p* < 0.01; * *p* < 0.05; ** *p* < 0.01; **** *p* < 0.0001.

**Figure 4 vaccines-12-00982-f004:**
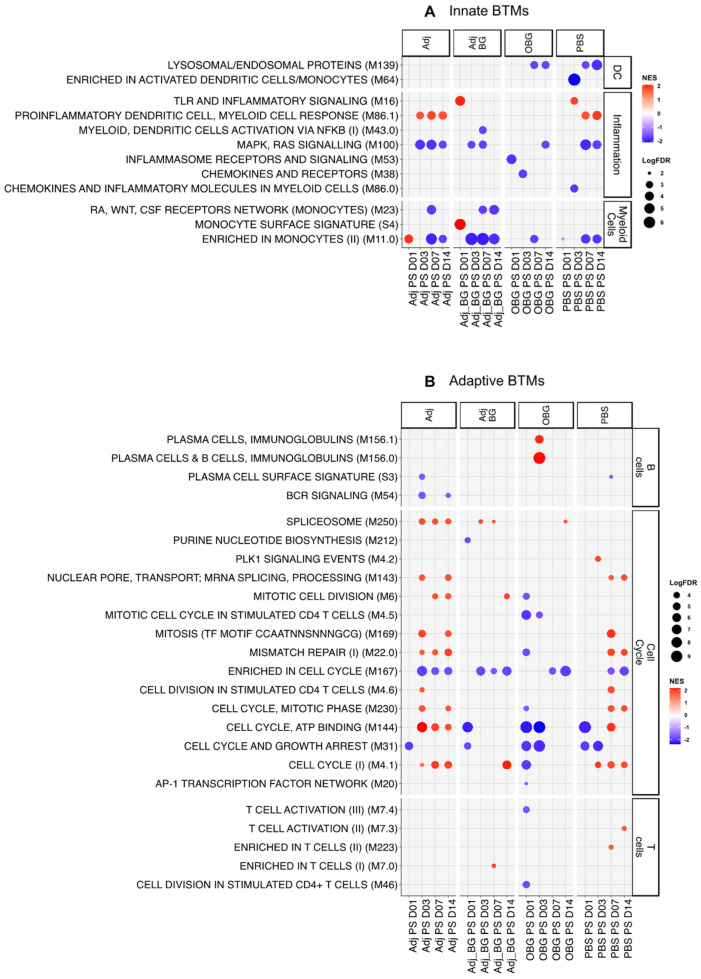
Transcriptomic profiles of PBMCs after stimulation. Pairwise comparisons of each group by itself against the baseline (D0) were performed and pre-ranked DEGs were subjected to GSEA using BTM as gene sets. Significant BTMs (FDR ≤ 0.05) were visualized using ggplot2. (**A**) Innate BTMs and (**B**) adaptive BTMs after stimulation.

**Figure 5 vaccines-12-00982-f005:**
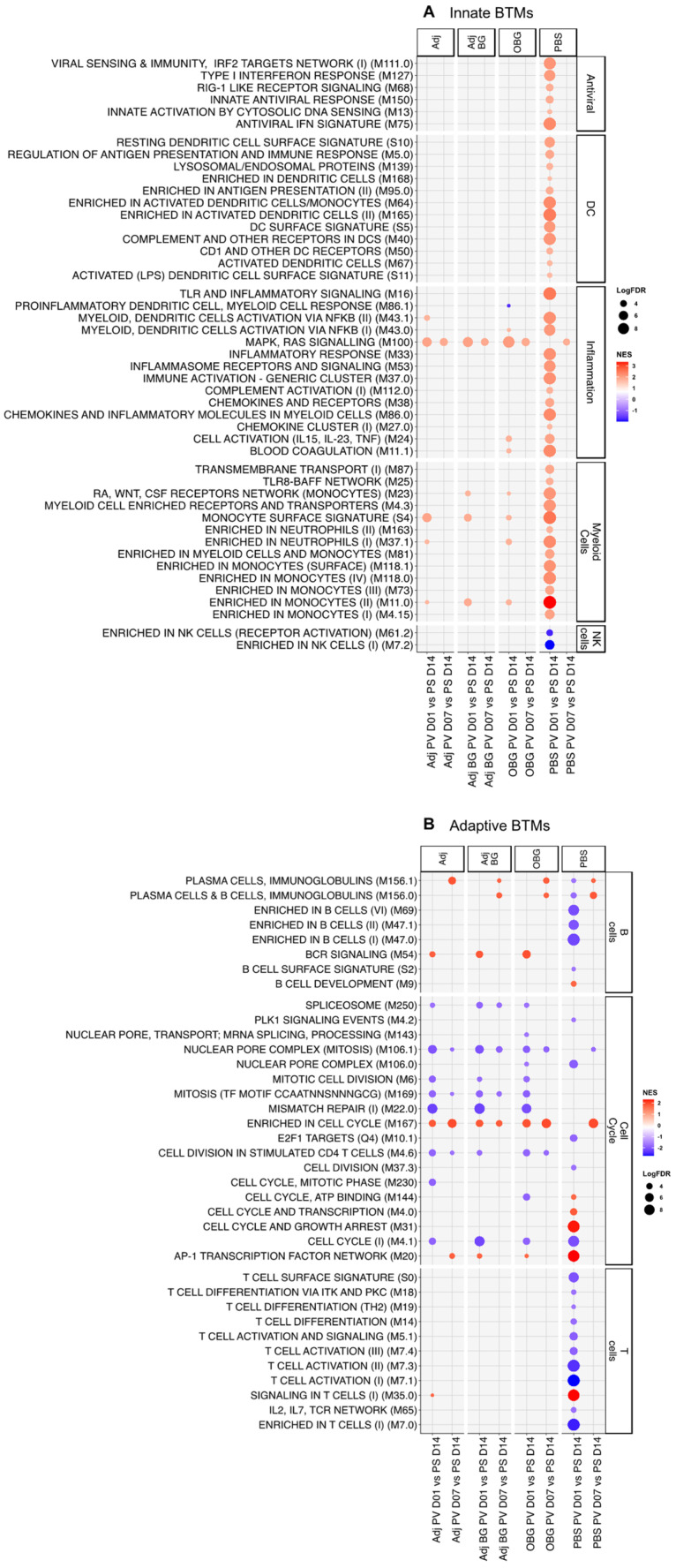
Transcriptomic profiles of PBMCs after vaccination. Pairwise comparison of each group by itself against the baseline (D14_PS) was performed and pre-ranked DEGs were subjected to GSEA using BTM as gene sets. Significant BTMs (FDR ≤ 0.05) were visualized using ggplot2. (**A**) Innate BTMs and (**B**) adaptive BTMs after vaccination.

**Table 1 vaccines-12-00982-t001:** The scoring criteria for monitoring injection site score.

Score		Description
0	Normal	no more than a visible injection site less than about 0.5 diameter zone of redness surrounding injection site
1	Mild	0.5–2 cm diameter discoloration no distinct palpable swelling may be irritation (occasional rubbing at injection site)
2	Moderate	2–5 cm diameter discoloration (or) palpable swelling may be irritation (persistent rubbing at injection site)
3	Sever	>5 cm diameter discoloration (and) visible and palpable swelling irritation and pain (persistent rubbing at injection site, withdrawal and vocalization upon palpation may be abscess, exudate

**Table 2 vaccines-12-00982-t002:** Number of DEGs (*p*. adjusted value ≤ 0.05) by comparing each treatment against its baseline.

Treatment	PBS	Adj	Adj_BG	OBG
Post-stimulation				
D1_PS	978	1045	1376	2617
D3_PS	703	6034	3771	3263
D7_PS	5939	5057	4149	4153
D14_PS	5008	5412	3891	4001
Post-vaccination				
D1_PV	1471	5119	4377	5238
D7_PV	3934	3972	3650	4222

## Data Availability

The sequencing data is available from the EMBL-EBI European Nucleotide Archive with accession numbers PRJEB79230 and ERP163417.

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
