# Peer review of "Impact of Oil-in-Water Adjuvanted β-Glucan on Innate Immune Memory in Piglets"

_vaccines, 2024, doi:10.3390/vaccines12090982_

Round 1
Reviewer 1 Report
Comments and Suggestions for Authors
This paper clearly reported the adjuvant effect of β-Glucan on the innate trained immunity of piglets through the leukocyte transcriptome, proinflammatory cytokines and clinical scoring after a first “training” and a second “stimulation” phase. The induction of innate immunity was analyzed during a “stimulation” by an LPS-adjuvanted Mycoplasma hypermania vaccine one day after weaning. Obviously, the authors failed to detect any meaningful trained innate immune memory reaction provoked by β-Glucan as expectation; Moreover, after vaccination, the expected early innate BTMs were only induced by the PBS control group. Therefore, the ß-glucans were unable to raise innate immunity but rather appeared to elicit innate immune tolerance. Anyway, their results about the failure of the β-Glucan to induce innate immune memory is helpful for further exploration to develop positive adjuvant for piglet vaccination. This manuscript is acceptable for publication after minor revision.
1. Piglet should be added as a keyword to precisely reflect the research content of this article.
2. The present manuscript just included the result of TNF and IL-1 ß to mirror the innate immune reaction, which is not enough for to the intact reflection of comprehensive innate immune responses after the first stimulation and second vaccination. More innate indexes, such as the function and number of macrophages and dendritic cells, are needed to better reflect the overall changes of innate immunity.
3. The exact definition of innate immune memory or trained innate immunity could be added in the discussion or introduction section to clarify the potential obscurity for the impact of β-Glucans used in their experiments.
Reviewer 2 Report
Comments and Suggestions for Authors
The study investigated the effects of β-glucans on piglet immune response. The study is relatively new in pigs though concept has been widely accepted in humans and mice. The evaluating method was just based on two cytokines and only one time point (Day 21 PV) of antibody response. The antibody production is a dynamic process including onset, peak and duration. One time point probably missed the true biological effects of β-glucans. Additionally, RNA-seq data were not compared among the treatments, which are more meaningful than comparisons among Days PS or PV. The limitations of the experimental design should be pointed out in the discussion to avoid misleading/misunderstanding. RNA-seq may not be the best technique for trained immunity study.
It is helpful to include how β-glucans works at the molecular level in the introduction.
Specific suggestions:
Line 21: change “which” to “that”.
Line 24: change " “ to “ “.
Line 89: culture time (hours or days)?
Line 116: change “Group of six piglets” to “Three groups (six piglets per group)”.
Line 127: change “age, and the following day received” to “age and, in the following day, received”.
Lines 136-137: blood volume ?
Line 145: change “of, TNF” to “of TNF”.
Line 152: change “temperature the plate” to “temperature, the plate”.
Line 165: the numbers should be expressed as such 3,969,430.
Line 170: the number of 347 should be expressed as Three hundred …
Line 181: A paired t-test is used when the difference between two variables for the same subject is tested. Paired test may not be appropriate for some between-groups tests.
Line 219: suggest changing “enhanced” to “increased or raised”.
Figure 1: NS, NS/NS, NS/LPS ? Figure 1F spelling error.
Table 2 needs a bottom line.
Comments on the Quality of English LanguageThe manuscript was well prepared.
Reviewer 3 Report
Comments and Suggestions for Authors
The authors evaluated the effect of adjuvanted-β-Glucan to induce innate immune memory in 14-day-old piglets by immunological as well as by BTM. Although the authors got negative results, I think this paper should be published expecting to enhance researches in this interesting concept. Specific comments follow.
Major points:
1. Line 74: The age and sex of the source of PBMCs are not clear. Have the authors collected them from new born piglets? The reactivity of the cells against β-Glucan must be different with age and/or sex.
2. Figure 1: Please explain “NS”.
3. Figure 2: I don’t see any stats. “ns, not significant; * p<0.05; ** p<0.01; ***p<0.001”.
4.
Minor points:
1. Line 84: “CO2” 2 should be subscript.
2. Line 138: “NaHCO3” 3 should be subscript.
Round 2
Reviewer 2 Report
Comments and Suggestions for Authors
Comments and suggestions have been adressed.